# Comparison of Two Tuberculosis Infection Tests in a South American Tertiary Hospital: STANDARD F TB-Feron FIA vs. QIAreach^TM^ QuantiFERON-TB

**DOI:** 10.3390/diagnostics13061162

**Published:** 2023-03-18

**Authors:** Gustavo Saint-Pierre, Daniel Conei, Patricia Cantillana, Mariella Raijmakers, Andrea Vera, Daniela Gutiérrez, Cristopher Kennedy, Paulina Peralta, Paulina Ramonda

**Affiliations:** 1Unidad Microbiología Sección Koch, Hospital Barros Luco Trudeau, Servicio de Salud Metropolitano Sur, Santiago 8900000, Chile; 2Programa de Virología, ICBM, Facultad de Medicina, Universidad de Chile, Independencia 1027, Santiago 8380453, Chile; 3Departamento de Ciencias de la Salud, Universidad de Aysén, Coyhaique 5951537, Chile; daniel.conei@uaysen.cl; 4Medicina Interna, University of Santiago, Santiago 9170022, Chile; 5Departamento de Medicina Interna, Hospital Barros Luco Trudeau, Servicio de Salud Metropolitano Sur, Santiago 8900000, Chile; 6Policlínico Infectología/Inmunología, Hospital Barros Luco Trudeau, Servicio de Salud Metropolitano Sur, Santiago 8900000, Chile; 7Servicio de Salud Metropolitano Sur, Santiago 8900000, Chile

**Keywords:** latent tuberculoses, latent tuberculosis infections, Koch’s disease, interferon-gamma release assay, tuberculin tests, rapid diagnostic test

## Abstract

Introduction: Tuberculosis (TB) is one of the most prevalent respiratory diseases in the world. In 2020 there were at least 9.9 million new infections, with 1.5 million deaths. Approximately 10% of people infected with *Mycobacterium tuberculosis* develop the disease during the first 2 to 5 years after infection. In South America, the diagnosis of Latent Tuberculosis Infections (LTBI) continues to be performed through the Mantoux tuberculin skin test (TST). Objective: The objective of our study was to compare the sensitivity of a new immunofluorescence IGRA test against a widely available IGRA kit on the market. Material and method: Close contact with infectious TB patients, HIV patients, or immunocompromised for another cause were recruited. Two interferon-gamma release assay (IGRA) diagnostic kits were used and compared with TST. Results: 76 patients were recruited, 93.42% were Chilean nationality, and 98.68% of the patients did not have immunosuppression. The sensitivity of the new technique was 88.89%, and the specificity was 92.50% in the study population compared to the IGRA previously used. In the subgroup older than 36 years, the sensitivity was 95.65%, and the specificity was 89.47%. Conclusion: IGRA techniques are a new resource in clinical laboratories to make an accurate diagnosis of LTBI in the region of the Americas. In our population, the greatest benefit of this new IGRA would be observed in people over 36 years of age, where the sensitivity of the technique was like that of the currently available test.

## 1. Introduction

Tuberculosis (TB) is an infectious disease that frequently produces respiratory symptoms and is caused by *Mycobacterium tuberculosis* (MTB) [1]. This infection is transmitted from person to person through airborne droplets expelled by patients with active lung disease (baciliferous) [2]. MTB infection is usually asymptomatic in healthy people [2], but it can also be presented as an active disease, the most characteristic symptom of coughing. It’s also frequent to observe mucopurulent sputum and sometimes hemoptysis, chest pain, weakness, weight loss, fever, and night sweats [3]. In 2020, there were at least 9.9 million new infections, with 1.5 million deaths, particularly the most affected group were patients living with HIV (214,000 deaths). In Latin America, PAHO reports estimated 291,000 cases, with 27,000 deaths. The multidrug-resistant TB (MDR-TB) diagnosis was made in 4007 cases, but only 89% of these started anti-tuberculosis treatment [4,5].

Approximately 5 to 10% of people infected with *Mycobacterium tuberculosis* develop the disease during the first 2 to 5 years after infection [6]. Patients who do not develop the disease are postulated to be due to a series of pathogen factors and host mechanisms, including the innate immune response that would eliminate the infection without leaving a trace of an immune response (resistance to tuberculosis infection) [6]. In other cases, patients would develop a state of persistent immune response to MTB antigens without clinical evidence of active disease [7]. From this finding of antigenic persistence, it is considered that a quarter of the world population could be infected with *M. tuberculosis* [8]. Individuals are carriers of TB infection, known as latent tuberculosis infection (LTBI) if bacterial clearance is achieved, becoming a potential reservoir for active tuberculosis [6].

Currently, in South America, the diagnosis of LTBI continues to be made through the tuberculin skin test (PPD in Spanish), also known as the Mantoux tuberculin skin test (TST) [9,10]. However, in the region, a high percentage of the newborn population is vaccinated against Bacillus Calmette–Guérin (BCG) for the prevention of complicated tuberculosis (meningitis TB or miliary TB) [11]. According to what was reported by Ynozente-Lázares et al. among the countries that 2017 had the highest coverage were Brazil (100%), Venezuela (100%), Argentina (97%), and Guyana (97%), while those with the lowest indicators are Peru (84%) and Paraguay (84%) [12]. In Chile, in 2021, coverage reached 98.4%, according to reports from the Ministry of Health. (Preliminary Report on National Immunization Coverage 2021, Period January–December 2021). However, BCG vaccination interferes with the diagnosis of LTBI through TST. This diagnostic technique presents an immunological cross-reactivity between antibodies against the vaccine and antibodies associated with exposure to MTB naturally, so in childhood and adolescence, the use of other diagnostic tests complementary to the TST in suspicion of LTBI [13] would be recommended. Likewise, according to what was stated by Zellweger et al., the TST diagnostic test is the oldest that evaluates LTBI. Still, it can be associated with various technical errors, has limited positive predictive value, is influenced by BCG vaccine, and various host conditions can reduce skin reactivity [14]. Therefore, current scientific studies have suggested evaluating other diagnostic techniques, such as interferon-gamma release assays (IGRA), since these have a higher specificity and are not affected by *M. bovis,* BCG vaccine, and other environmental mycobacteria. IGRAs use only two M. tuberculosis-specific antigens (ESAT−6 and CFP−10) that are absent from *M. bovis* and BCG but present in some environmental Mycobacterium spp. These antigens stimulate the lymphocytes after an incubation process at 37 °C, generating interferon-gamma, which is only released and detected by diagnostic techniques and only if the patients had previously been exposed to these antigens by contact during their life with *M. tuberculosis*, currently presenting with LTBI [14]. In the case of the skin test, the TST IC−65 technique is currently used, which was introduced in Romania in 1965, and it has been used as an “in vivo” assay for the surveillance of *M. tuberculosis* infection (diagnosis of LTBI), the diagnostic kit includes 54 proteins, which were identified through mass spectrometry, among them MPT64, ESAT 6 and CFP 10, antigens recognized in MTB and others that have cross-reactivity against other non-TB Mycobacteria [15].

No South American reports compare diagnostic tests for the study of LTBI in highly complex hospitals. Our laboratory group decided to evaluate a new diagnostic test available in the southern cone. The objective of our study was to compare a new immunofluorescence IGRA test against a commercially available IGRA kit, which had previously been compared against other techniques considered the gold standard for the determination of interferon in the blood of patients suspected of LTBI. The main difference between the included reagents is that the new diagnostic kit consists of a third MTB-specific antigen, TB7.7 [16].

## 2. Materials and Methods

### 2.1. Study Design and Participants

This prospective and longitudinal study was conducted in the Infectious Diseases/Immunology Department of the Hospital Barros Luco Trudeau, Santiago, Chile, a tertiary care center located in the south of the country’s capital, with an assigned population of 1,500,000 users. The study was approved by the ethics committee of “Servicio de Salud Metropolitano Sur”, Santiago, and written informed consent was obtained from all study participants.

For this study, subjects classified as close contacts of recently diagnosed (less than 3 months) positive pulmonary TB patients, HIV patients, or who had to receive immunosuppressive drug treatment for an underlying medical condition, particularly methotrexate therapy or biologics, were recruited. Household contacts of patients with pulmonary TB were defined as a large group of family members residing with the pulmonary TB index case in the same household for >3 months and having a common kitchen arrangement or sleeping together in the same bedroom. Subjects with a history of previously treated TB; HIV; positivity for hepatitis B and C; pregnant women were not excluded from the study. Patients were recruited from September to December 2022.

Inclusion criteria for the group of TST-positive subjects: Positive test greater than 10 mm, being between 18 and 65 years of age, not having been previously diagnosed with TB without treatment, or being under treatment for TB during the IGRA sample collection.

A control group was used with 20 users or hospital employees with a low risk of latent tuberculosis, who was asked for a tuberculin test and whose result was negative or ≤9 mm, who were between 18 and 65 years old, had not previously been diagnosed by TB or being treated for TB.

When we took the TST, all study participants had 9 mL of blood drawn (3 mL in 2 tubes with lithium heparin and a 3 mL tube with EDTA) for IGRA studies and complementary tests (CBC). Complete with white blood cell count). Clinical and demographic data collected from patients included age, gender, and key parameters of white blood cell counts from the complete blood count. Appendix A.

### 2.2. Standard F TB-Feron FIA Assay

For each participant, 8 mL of blood was obtained and transferred to 2 tubes with lithium heparin. In less than 4 h, the volume was transferred to Standard E TB-Feron Tubes (Nil, TB Ag, and the mitogen tube). The antigens evaluated were ESAT−6, CFP−10, and TB7.7 (This last one is specific for TB and available only in this commercial technique). After collection, the tubes were incubated at 37 °C for 16 to 24 h. IFN-γ quantification was processed with the STANDARD F2400 equipment (SD Biosensor, Gyeonggi-do, Republic of Korea), an automated FIA (fluorescent immunoassay) device. From sample loading into the cartridge to fluorescence intensity reading and IFN-γ quantification, followed by interpretation, the test is performed automatically within 15 min. Results are interpreted according to the manufacturer’s instructions [17,18].

### 2.3. QIAreach QFT Assay

The other previously collected lithium heparin tube was used to draw 1 mL of whole blood from all study participants for transfer directly into the QIAreach QFT Blood Collection Tube (equivalent to QFT-Plus ESAT−6 Antigens tube TB2, CFP−10) and incubated at 37 °C for 16–24 h. Plasma samples were centrifuged at 3000× *g* for 15 min and analyzed according to the manufacturer’s instructions.

First, a charged eHub was powered on, and the eStick was inserted into the eHub port. Once connected and powered on, the eHub was in ready mode. A total of 150 µL of diluent was added to the processing tube. Next, 150 µL of plasma sample was transferred to the same processing tube. The resulting solution was mixed by pipetting up and down at least 8 times. Finally, 150 μL of this mixture was aliquoted from the processing tube into the sample port of the inserted eStick. The trial started automatically, and the status was displayed on the eHub when the mix was detected. At the end of the trial, the test result (+ or −) and the time to result (TTR) were indicated in the eHub screen.

### 2.4. Statistical Analysis

Data were statistically analyzed using GraphPad Prism 9.3.1 and presented as No (%) or median (interquartile range) unless otherwise specified. Sensitivity, specificity, and overall agreement (proportion of overall true results) in the Standard F technique were calculated using the QIAreach QFT as the reference standard. A subanalysis was also performed on samples from immunocompromised patients (less than 500 lymphocytes per uL).

Ethical: authorized by the ethics committee of “Servicio de Salud Metropolitano Sur” no. 22/2022.

## 3. Results

In total, 76 patients referred for a TST study were analyzed to rule out the suspicion of LTBI. In the sample, the median age was 38 years. (IQR 22–54). There were 73.68% women, and 93.42% were Chileans. Of the patients, 98.68% did not have immunosuppression secondary to HIV (75/76). Additionally, 98.68% were referred due to a requirement to use biological drugs/methotrexate or in a contact study for a positive TB patient in their residence (Table 1).

Table 2 summarizes the comparison between the new IGRA technique through immunofluorescence diagnosis. This was compared against the QIAreach technique, which previous studies had compared against the current gold standard (GS), the Quantiferon^®^ TB Gold technique. Because we had the QIAreach technique available to be able to use it in this investigation, Fukushima et al. compared it against Quantiferon^®^ TB Gold [16].

Table 2(A) shows the sensitivity of STANDARD F TB-Feron FIA. The sensitivity of the new technique was 88.89%, and the specificity was 92.50% in the study population. Since TST was also performed at the hospital, it was decided to compare both diagnostic techniques against this one. Table 2(C) shows that STANDARD F TB-Feron FIA had an overall sensitivity of 82.86% and a specificity of 80.65%, having an even better specificity than the comparator pattern of 62.50%.

An analysis by age subgroups was also performed. Separating the samples into a group Under 35 years (34 samples) and greater than or equal to 36 years (42 samples). Table 3 (A) showed a sensitivity of 76.92% and a specificity of 95.24%. In Table 4 (A), in those over 36 years of age, the sensitivity of the new technique was 95.65% and the specificity 89.47%.

Table 5 shows the median number of leukocytes, lymphocytes, and gamma interferon release caused by the antigens in the tube of the new IGRA technique under evaluation. The median interferon release for the positive TST was 0.46 and 0 in the case of the negative TST. Finally, in the case of TST-positive patients, from the subgroup positive for the new IGRA technique, a median of 1.33 ug/mL of gamma interferon was observed when stimulated by the technique’s antigens (Data not shown. Information available in Appendix A).

## 4. Discussion

In 2022, the Chilean Ministry of Health generated an update on tuberculosis management standards [19]. This regulation dictates the action in the country for the diagnosis and treatment of all positive cases of TB, both in public and private hospitals. For the first time, regulation of the Ministry of Health in Chile considers the study and treatment for LTBI, which is a paradigm shift for the national TB program, as the diagnosis must be through TST or IGRA. The new standard considers that the TST is reactive when the induration is 10 mm or more. In the case of patients with HIV and other immunosuppressed patients, a TST of 5 mm or more is considered reactive. Reactive TST or positive IGRAs should not be repeated, as a result will persist over time even after LTBI treatment has been performed. These tests cannot differentiate LTBI from tuberculosis disease, so a good clinical and imaging diagnosis is key in case of suspicion of TB disease. This standard also dictates that patients diagnosed with LTBI should receive treatment with rifapentine + isoniazid once a week for 12 weeks. [19]. ITBL study is important for Chile since adequate treatment should be given to patients suspected of LTBI, particularly HIV patients, immunosuppressed by biological drugs or close contact of positive TB patients with a diagnosis of fewer than 2 years since they are candidates to become ill with Tuberculosis within the next 5 years [6].

Due to this background, it was decided to implement a new diagnostic technique in a highly complex hospital in the southern area of the capital of Chile. The registered population in the aforementioned area, which is more than 1.5 million inhabitants, with a TB incidence of about 19.8 per 100,000 persons, is higher than other regions of Chile (13.3 per 100,000 persons) [20]. As a research team, we considered that it was necessary to perform an IGRA technique beyond the TST. IGRA techniques such as TB-Gold and T-SPOT were not easy to implement in our institution due to the requirements of exclusive technicians for their implementation. Unlike point-of-care equipment, it is easy to load, such as those evaluated in this study, which allows more than 24 patients to be analyzed within an hour.

The first comparison made in the study was between Standard F TB-Feron FIA and QIAreach™. Table 2 summarizes the comparison with the novel IGRA technique, and Table 2(A) shows the sensitivity of Standard F TB-Feron FIA. The sensitivity of the new technique was 88.89%, and the specificity was 92.50% in the study population. In the population under study, the new technique would provide valid results for the diagnosis, and these results would give greater certainty than the TST in the face of clinical suspicion of LTBI or in the case that it is necessary to rule out LTBI to start immunosuppressive treatment. In Chile, with positive results for LTBI, action must be taken following current regulations. Therefore, 12 weeks must be treated with anti-tuberculosis drugs. Due to the above, it is necessary to rule out false positive cases. That is, adequate specificity in the new diagnostic technique would ensure that anti-tuberculosis treatments are not indicated since antimicrobial drugs in prolonged use could generate unwanted adverse drug reactions and alter the intestinal microbiota. Table 2(C) shows that Standard F TB-Feron FIA had an overall sensitivity of 82.86% and a specificity of 80.65%, having an even better specificity than the comparator standard 62.50% (QIAreach versus TST). However, the results compared against TST may have been influenced by probable false positive cases in the population under 36 years of age, in whom there could still be an immune response to BCG vaccination (until 2004, there were two vaccination doses, the first within the first month of the newborn and then a second dose at 6 years) [21]. In countries without BCG vaccination, the adolescent and young adult population would not have recognition of antigens against vaccination in the TST test, and, therefore, negative results would not have BCG interference. In the case of our study, this could explain the decreased sensitivity (45% in those under 35 years of age vs. 80% in those over 36 years of age when comparing Standard F TB-Feron FIA vs. TST).

Given this background, it was decided to analyze the data by age subgroups. Separating the samples into a group under 35 years (34 samples) and a group greater than or equal to 36 years (42 samples). In Table 3(A) (Population under 36 years of age), a sensitivity of 76.92% was observed, lower than that observed in the total number of cases, and a specificity of 95.24%. In Table 4(A), for 36 years or older, the sensitivity of the new technique was 95.65% and the specificity 89.47%. In this case, it was the greatest success of the new technique when compared to the GS. In Table 4(C), the sensitivity of the new technique is 80.00%, and the specificity of 76.47% compared to TST. This increase in sensitivity is probably due to the decrease in TST-positive cases in this subgroup due to the recognition of BCG vaccination antigens in the first group analyzed (under 36 years of age). In Table 5, the median of the positive TST group is greater than that of the negative TST group. The median of the positive group was 0.46, a result lower than expected. Therefore, we decided to evaluate the median of the positive TST subgroup Standard F TB-Feron FIA with a positive result, whose median was 1.33 ug/mL. This result was unexpected for our group since values greater than 3.39 ug/mL (median) were expected according to the reports described in the literature [22]. It should be noted that in values close to the median 0.46 ug/mL for the antigen tube in the population studied, if the result is negative or indeterminate due to the equation between mitogen and Nil, it is suggested to complement the results with TST or repeat the test in patients at risk of LTBI, such as HIV, other immunosuppression, use of biological drugs, close contact with a positive patient, to avoid false negative results. To resolve this dilemma, we consider it necessary to carry out similar studies with other populations worldwide in places with different prevalences of TB. Furthermore, suppose the determination of using tests that detect results qualitatively was made in the hospital service. In that case, carrying out a second quantitative test against negative cases in patients with LTBI risk factors is strongly suggested.

As authors, we believe that the new diagnostic technique recently available in South America brings benefits for patients with LTBI. First, the lower cost compared to other IGRAs. In Chile, the current cost is 25 USD per exam. In addition, the technique is simple to implement in medical laboratories.

Limitations: The study was in a single center in Santiago de Chile. There was only one HIV patient, so it could not be compared with non-HIV immunosuppressed patients. All the patients studied had BCG in the newborn period (within the national immunization program since 1949).

## 5. Conclusions

IGRA techniques are a new resource in clinical laboratories for the diagnosis of LTBI in South America, all with a view to eliminating Tuberculosis cases by 2035. In our population, the greatest benefit of this new IGRA available in the laboratory would be observed in people over 36 years of age, where the sensitivity of the technique was like that of GS and with excellent specificity (95.65% and specificity 89.47%). We invite other groups of researchers to publish their data in countries where there is no influence of BCG vaccination, which can interfere with diagnosis through TST, as occurs in Chile and South America.

## Figures and Tables

**Table 1 diagnostics-13-01162-t001:** Demographic description of patients studied for TST in the Infectious Diseases/Immunology department. (September to December 2022). (*n* = 76).

		TST	
	Total	Positive	Negative
Age (average)	41.42	42.66	31
Age (median)	38.00	42	36
IQR	22–54	32–55	31–49
Sex (female)	73.68%	60.71%	39.29%
Nationality (chilean)	93.42%	93.33%	93.55%
Foreign	6.58%	6.67%	6.45%

**Table 2 diagnostics-13-01162-t002:** (A). Comparison between the new IGRA technique and the gold standard. Samples collected from HBLT patients. (September to December 2022). (B). Comparison between the new IGRA technique and the gold standard. Samples were collected from HBLT patients (September to December 2022). (*n* = 76). (C). Comparison between the new IGRA technique and the gold standard. Samples collected from HBLT patients (September to December 2022). (*n* = 76).

(A)				
		GOLD STANDARD QIAREACH		
		Positive	Negative	Total
SD Biosensor	Positive	32	3	35
	Negative	4	37	41
	Total	36	40	76
**(B)**				
		**TST**		
		**Positive**	**Negative**	**Total**
E okQIAreach	Positive	30	15	45
	Negative	6	25	31
	Total	36	40	76
**(C)**				
			**TST**	
		**Positive**	**Negative**	**Total**
SD Biosensor	Positive	29	6	35
	Negative	6	25	31
	Total	35	31	66

Sensitivity (new technique) 88.89%; Specificity (new technique) 92.50%. Sensitivity (new technique) 83.33%; Specificity (new technique) 62.50%. Sensitivity (new technique) 82.86%; Specificity (new technique) 80.65%.

**Table 3 diagnostics-13-01162-t003:** (A). Comparison between the new IGRA technique and the gold standard. Samples collected from HBLT patients under 36 (September to December 2022). (*n* = 34). (B). Comparison between QIAreach and TST. Samples collected from HBLT patients under 36 years of age (September to December 2022) (*n* = 34). (C). Comparison between the new IGRA technique (SD BIOSENSOR) and TST. Samples collected from HBLT patients under 36 years of age, (September to December 2022) (*n* = 34).

(A)				
		GS QIAREACH		
		Positive	Negative	Total
SD Biosensor	Positive	10	1	11
	Negative	3	20	23
	Total	13	21	34
**(B)**				
		**TST**		
		**Positive**	**Negative**	**Total**
QIAreach	Positive	10	3	13
	Negative	10	11	21
	Total	20	14	34
**(C)**				
		**TST**		
		**Positive**	**Negative**	**Total**
SD Biosensor	Positive	9	2	11
	Negative	11	12	23
	Total	20	14	34

Sensitivity (new technique) 76.92%; Specificity (new technique) 95.24%. Sensitivity (new technique) 50.00%; Specificity (new technique) 78.57%. Sensitivity (new technique) 45.00%; Specificity (new technique) 85.71%.

**Table 4 diagnostics-13-01162-t004:** (A). Comparison between the new IGRA technique and the Gold standard. Samples collected from HBLT patients ≥ 36 years of age, (September to December 2022) (*n* = 42). (B). Comparison between QIAreach and TST. Samples collected from HBLT patients > = 36 year of age (September to December 2022) (*n* = 42). (C). Comparison between the new IGRA technique (SD BIOSENSOR) and TST. Samples collected from HBLT patients ≥ 36 year of age (September to December 2022) (*n* = 42).

(A)				
		GS QIAREACH		
		Positive	Negative	Total
SD Biosensor	Positive	22	2	24
	Negative	1	17	18
	Total	23	19	42
**(B)**				
		**TST**		
		**Positive**	**Negative**	**Total**
QIAreach	Positive	20	3	23
	Negative	5	14	19
	Total	25	17	42
**(C)**				
		**TST**		
		**Positive**	**Negative**	**Total**
SD Biosensor	Positive	20	4	24
	Negative	5	13	18
	Total	25	17	42

Sensitivity (new technique) 80.00%; Specificity (new technique) 82.35%. Sensitivity (new technique) 95.65%; Specificity (new technique) 89.47%. Sensitivity (new technique) 80.00%; Specificity (new technique) 76.47%.

**Table 5 diagnostics-13-01162-t005:** Evaluation of gamma interferon levels stimulated by antigens using the novel IGRA technique in patients being studied in the infectious disease department with the TST technique (positive and negative cases).

TST	TST (mm) (Median)	Leukocytes ug/mL (Median)	Leucocitos (ug/mL) (Median)	Ag SD Biosensor Tube ug/mL (Median)
Positive	14	7250	2240.375	0.46
Negative	0	5645	1595.85	0

## Data Availability

All our research data are available in Appendix A.

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
