# Peer review of "Comparison of Two Tuberculosis Infection Tests in a South American Tertiary Hospital: STANDARD F TB-Feron FIA vs. QIAreachTM QuantiFERON-TB"

_diagnostics, 2023, doi:10.3390/diagnostics13061162_

Round 1

Reviewer 1 Report

Minor revision required

Author Response

Dear Reviewer:

We appreciate the excellent review you have given us of our scientific article. We have made all the corrections indicated. We believe that thanks to them our work has improved even more. We hope that with the new version that will be uploaded to the platform we will be able to comply with your instructions.

Kind regards

Reviewer 2 Report

The manuscript entitled “First clinical comparison of STANDARD F TB-Feron FIA and the QIAreachTM QuantiFERON-TB for tuberculosis infection in Tertiary Public Hospital in South America” is good and can be acceptable for publication.

What author think about the revised title “Comparison of two tuberculosis infection tests in a South American tertiary hospital: STANDARD F TB-Feron FIA vs. QIAreachTM QuantiFERON-TB." It is not compulsory. If the authors think it is suitable, they can opt it.

In the line 23, Approximately 22 10% of people infected with Mycobacterium tuberculosis (LTBI). I think LTBI stands for latent TB infection.

In the results section, it is important to clearly state which techniques or interventions were compared and how the comparisons were made. This can be done by including subsections or headings that describe the different analyses or comparisons performed.

The discussion needs to be conclusive. It is not clear which test was better in term of sensitivity and specificity. Based on the findings, what are the suggestion and recommendations for the diagnosis of LTBI?

Author Response

Reviewer 2:

Point-by-point response:

Dear reviewer, I appreciate your comments. We have accepted his suggestions. The responses to your suggestions are directly described in this document.

Attached word with corrections from all reviewers.

Kind regards.

What author think about the revised title “Comparison of two tuberculosis infection tests in a South American tertiary hospital: STANDARD F TB-Feron FIA vs. QIAreachTM QuantiFERON-TB." It is not compulsory. If the authors think it is suitable, they can opt it.  We accept the title change.

In the line 23, Approximately 22 10% of people infected with Mycobacterium tuberculosis (LTBI). I think LTBI stands for latent TB infection. we change it. 

In the results section, it is important to clearly state which techniques or interventions were compared and how the comparisons were made. This can be done by including subsections or headings that describe the different analyses or comparisons performed. 

We have accepted your proposal and have corrected the paragraphs in the results.

The discussion needs to be conclusive. It is not clear which test was better in term of sensitivity and specificity. Based on the findings, what are the suggestion and recommendations for the diagnosis of LTBI?

We have added a paragraph with our conclusions on the recommendation of the use of this new diagnostic technique available in South America, due to cost and similar results to the comparator.

“As authors, we believe that the new diagnostic technique recently available in South America brings benefits for patients with LTBI. First, the lower cost compared to other IGRAs, in Chile the current cost is 25 USD per exam. In addition, the technique is simple to implement in medical laboratories”.

Reviewer 3 Report

In this work, the authors have developed a new technique to detect Mycobacterium tuberculosis in patients. Mycobacterium tuberculosis is a global threat killing ~ 2 million people annually. There has been increasing in the number of people infected with tuberculosis in South America. To make the situation worse the advent of multi-drug resistance strains and the ability of the pathogen to lie dormant for decades have made the detection of the pathogen challenging. Any new test/technique that can improve the sensitivity of detecting the pathogen will be of utmost importance.

Overall, the new test kit looks promising.

However, I’ve a couple of suggestions:

Line 23, 42, 53, 60, 85, 89, 92, 342, 355: Mycobacterium tuberculosis should be italicized.

Line 85: M. bovis should be italicized.

Line 91-92 looks like it is in a different font.

Tables 1-4 need formatting.

Author Response

Reviewer 3:

Point-by-point response:

Dear reviewer, I appreciate your comments. We have accepted his suggestions. The responses to your suggestions are directly described in this document.

Attached word with corrections from all reviewers.

In this work, the authors have developed a new technique to detect Mycobacterium tuberculosis in patients. Mycobacterium tuberculosis is a global threat killing ~ 2 million people annually. There has been increasing in the number of people infected with tuberculosis in South America. To make the situation worse the advent of multi-drug resistance strains and the ability of the pathogen to lie dormant for decades have made the detection of the pathogen challenging. Any new test/technique that can improve the sensitivity of detecting the pathogen will be of utmost importance.

Overall, the new test kit looks promising.

However, I’ve a couple of suggestions:

Line 23, 42, 53, 60, 85, 89, 92, 342, 355: Mycobacterium tuberculosis should be italicized. we correct it

Line 85: M. bovis should be italicized. we correct it

Line 91-92 looks like it is in a different font. we correct it

Tables 1-4 need formatting. we correct it.

Dear reviewer, we thank you for taking the time to correct our work. We hope the corrections will be satisfactory.

Kind regards.

Round 2

Reviewer 2 Report

The authors have incorporated the changes therefore can be accepted for publication